# A computational analysis of the oncogenic and anti-tumor immunity role of P4HA3 in human cancers

Hong Yan Huang[1]◉, Fu Wei Zhang[2]◉, Jie Yu[3]◉, Yan Hong Xiao[4,5], Di Zhu[1], XiaoLin Yi[1], XiaoHua Lin[1], Ming Jin[6], Hai Yun Jin[7]*, Yong Sheng Huang[8]*, Shu Wei Ren[4,5]*

1 Department of Breast Surgery, Zhujiang Hospital, Southern Medical University, Guangdong, China,
2 Department of Thoracic Surgery, Zhujiang Hospital, Southern Medical University, Guangdong, China,
3 Department of General Surgery, Zhujiang Hospital, Southern Medical University, Guangdong, China,
4 Department of Clinical Laboratory, The Sixth Affiliated Hospital, Sun Yat-sen University, Guangdong, China, 5 Biomedical Innovation Center, The Sixth Affiliated Hospital, Sun Yat-sen University, Guangdong, China, 6 Department of Gastroenterology, Shenzhen Hospital, Southern Medical University, Guangdong, China, 7 Department of Gynaecology and Obstetrics, Southern Hospital Taihe Branch, Southern Medical University, Guangdong, China, 8 Cellular & Molecular Diagnostics Center, Sun Yat-Sen Memorial Hospital, Sun Yat-Sen University, Guangdong, China

◉ These authors contributed equally to this work.
* shucai42521@126.com (HYJ); huangysh65@mail.sysu.edu.cn (YSH); renshw3@mail.sysu.edu.cn (SWR)

**Data Availability Statement:** All relevant data are in the manuscript and its Supporting information files.

## Abstract

Prolyl-4-hydroxylase subunit alpha3 (P4HA3) is a triple helical procollagen synthesis protein. The role of P4HA3 in cancer development is not well known and lacks comprehensive analyses among human cancers. This study aimed to investigate the relationship between P4HA3 expression and anti-tumor immunity and its prognostic value in pan-cancer. P4HA3 expression was analyzed from TIMER2.0, GTEx, GEPIA2.0 and TCGA databases. Genetic and DNA methylation alterations, survival analysis and proteins co-expression analysis of P4HA3 in cBio Cancer Genomics Portal, TCGA, GSCA and TIMER2.0. The correlation between P4HA3 expression and immune infiltration was analyzed by TIDE, XCELL, MCPCOUNTER, and EPIC. We performed EdU and transwell experiments to evaluate the influence of P4HA3 on the proliferation, migration and invasion abilities of different tumors. Patients derived xenograft (PDX) and subcutaneous transplantation models were utilized to explore the correlation between P4HA3 and immunotherapy response in triple-negative breast cancer (TNBC). Among 33 types of cancers, P4HA3 had generally different expression between different tumors, further analysis showed that the expression of P4HA3 was correlated with the cells infiltration of the tumor microenvironment (TME). The expression of P4HA3 was positively with the cell proliferation markers and epithelial-mesenchymal transition (EMT) markers. Moreover, P4HA3 deficiency inhibited the proliferation, migration and invasion abilities of tumor cells, and promoted anti-tumor immunotherapy of PD-1/PD-L1 inhibitor. This pan-cancer analysis of P4HA3 provides a comprehensive understanding of its oncogenic and prognosis role in different cancers, P4HA3 abnormal expression could be a useful biomarker for predicting the effectiveness of immunotherapy in cancer patients.

**Funding:** This study was supported by the Guangzhou Basic and Applied Basic Research Project (2023A04J2445 to HYH), Science and Technology Program of Guangzhou (202201010987 to JY), President Foundation of Zhujiang Hospital, Southern Medical University (yzjj2021qn18 to JY) and Guangdong Basic and Applied Basic Research Foundation (2021A1515111138 to YSH; 2022A1515110144 to JY), National Natural Science Foundation of China (82203703 to SWR). The funders had no role in study design, data collection and analysis, decision to publish, or preparation of the manuscript.

**Competing interests:** The authors have declared that no competing interests exist.

## Author summary

The results in this study presented the close correlation and the significant prognostic value of P4HA3 aberrant expression among diverse human tumors. Because the expression of P4HA3 was up-regulated in various of tumors and correlated with worse survival prognosis, we thought that P4HA3 possibly become a new therapeutic target for cancers. More importantly, our research provided a significant perspective in the important role of P4HA3 in anti-tumor immunotherapy and CAFs infiltration. The aberrant expression of P4HA3 might modulate tumor metabolic activity, immune cells and EMT processes in cancers. Future prospective research focusing on P4HA3 and TME could be useful for providing an immuno-based and metabolic-based anti-tumor therapeutic strategies.

## Introduction

Up to now, diagnosis and therapy of cancer have been greatly improved, but there will be about 30 million new cancer cases in 2040 according to the report of GLOBOCAN, and cancer remains a huge social problem [1]. Genetic variants and epigenetic regulation closely associated with the occurrence and development of tumors [2], TME is a multicellular system that comprise infiltrating immune cells, CAFs, and stromal cells, many therapeutic strategies targeting TME such as PD-1/PD-L1 inhibitor and immune vaccines have achieved great advances in treating cancers [3,4]. Thus, it is very meaningful to explore the function and mechanism of cancer-associated genes in cancer malignant development.

P4HA3 was firstly identified as an α-subunit of prolyl hydroxylase which is participated in triple helical procollagen synthesis [5]. Additionally, P4HA3 was shown to be high expression in obesity and contribute to the development of type 2 diabetes [6]. P4HA3 was significantly upregulated in gastric cancer (GC) [7], stomach adenocarcinoma [8], melanoma [9] and P4HA3 upregulation was involved in the metastasis and prognosis of GC patients [8]. However, the relationship between P4HA3 and TME in various of cancers is largely unknown.

In this study, we investigated the expression and function of P4HA3 based on the different datasets in various of cancers, including: adrenocortical carcinoma (ACC), bladder urothelial carcinoma (BLCA), breast cancer (BRCA), cervical squamous carcinoma (CESC), cholangiocarcinoma (CHOL), colon adenocarcinoma (COAD), lymphoid neoplasm diffuse large B-cell lymphoma (DLBC), esophageal carcinoma (ESCA), glioblastoma multiforme (GBM), head and neck squamous cell carcinoma (HNSC), kidney chromophobe (KICH), kidney renal clear cell carcinoma (KIRC), kidney renal papillary cell carcinoma (KIRP), acute myeloid leukemia (LAML), brain low grade glioma (LGG), liver hepatocellular carcinoma (LIHC), lung adenocarcinoma (LUAD), lung squamous carcinoma (LUSC), mesothelioma (MESO), ovarian serous cystadenocarcinoma (OV), pancreatic adenocarcinoma (PAAD), pheochromocytoma and paraganglioma (PCPG), prostate adenocarcinoma (PRAD), rectum adenocarcinoma (READ), sarcoma (SARC), skin cutaneous melanoma (SKCM), stomach adenocarcinoma (STAD), testicular germ cell tumor (TGCT), thyroid carcinoma (THCA), thymoma (THYM), uterine corpus endometrial carcinoma (UCEC), uterine carcinosarcoma (UCS) and uveal melanoma (UVM). We explored the relationship between P4HA3 expression profile and prognosis value, genetic alteration, epigenetic regulation and TME among different cancers. Additionally, we performed the functional role of P4HA3 in breast cancer, colon cancer and lung cancer *in vitro*. More importantly, we also explored the association between P4HA3 depleted and cancer immunotherapy in breast cancer *in vivo* experiment. Our study indicates

that P4HA3 depleted can promote anti-tumor immunity and P4HA3 upregulation maybe a potential prognostic molecular.

## Results

### Expression profiles and prognostic value of P4HA3 in different tumor types

To explore the clinical significance of P4HA3 among different tumors, we evaluated P4HA3 mRNA expression profile across TCGA and GTEx cancers by using the TIMER 2.0 and SangerBox tools. Compared to that normal tissues, P4HA3 mRNA expression was upregulated in BRCA, CHOL, COAD, ESCA, GBM, HNSC, KIRC, LUAD, LUSC, PCPG, PRAD, READ, STAD, THCA, ACC, LAML, PAAD, TCGT, and UCS tissues, suggesting that P4HA3 might function as an oncogenic role in the occurrence and development of diverse cancers (Fig 1A and 1B). Based on the GEPIA dataset, we found P4HA3 expression was positively correlated with advanced cancer stages in KIRC (P = 0.0118), LUAD (P = 0.0233), and STAD (P = 0.0396) (Fig 1C).

Next, we further analyzed the over survival (OS) and disease-free survival (DFS) of P4HA3 across the 33 different tumors in GEPIA datasets, the results were shown in Fig 1D and 1F. Elevated P4HA3 expression was correlated with poor OS in BLCA, CESC, COAD, KIRP, STAD, THCA, UCEC and UVM (P < 0.05) (Fig 1D), especially in COAD (P = 0.0026), KIRP (P = 0.0012), and STAD (P = 0.00068) (Fig 1E). Moreover, Elevated P4HA3 expression was correlated with poor DFS in COAD (P = 0.0051), KIRC (P = 0.0055) and LUAD (P = 0.0053) (Fig 1D and 1F). These results showed that elevated P4HA3 expression level was a great molecular significantly affecting the survival of different types of cancers.

### The genetic alteration and epigenetic regulation of P4HA3 among different tumors

Based on above results that P4HA3 is generally high expression in various of cancers, we explored the genetic alteration and epigenetic regulation of P4HA3 from different databases. We firstly explored the genetic alteration of P4HA3 in TCGA pan-cancer atlas dataset by using the cBioPortal online database, we found that the overall alteration frequency of P4HA3 was widely high in various of cancers, P4HA3 had the highest genetic alteration frequency in SKCM, ESCA, OV, UCEC, HNSCC and BRCA (Fig 2A). Then, we analyzed the mutation frequency of 23 cancers by using SangerBox tool, the data showed that P4HA3 have relatively low mutation frequency in various of cancers (Fig 2B). Because of the overexpression of P4HA3 in cancers, we analyzed the copy number amplification variation of the P4HA3 DNA fragment, and found a positive correlation between the copy number amplification variation and P4HA3 expression among various of cancers, especially in UCEC, HNSCC, LUAD, OV, KIRP, ESCA, SKCM and THYM (Fig 2C) (FDR < 0.05).

DNA methylation (DNAm) is an important epigenetic modification of chromosomes that playing a crucial role in gene regulation in cancers development [10,11]. We explored the relationship between DNA methylation and P4HA3 expression by using the gene set cancer analysis (GSCA) tool among different types of cancers. We found that there was negative correlation between DNA methylation and P4HA3 expression in cancers (Fig 2D), especially in HNSCC, COAD, OV, THYM, SKCM, KIRP and SARC (Fig 2E). The DNA methylation is regulated by methyltransferases (DNMTs) including DNMT1, DNMT2, DNMT3A and DNMT3B [12]. We found that the remarkable negative correlation with DNMTs and P4HA3

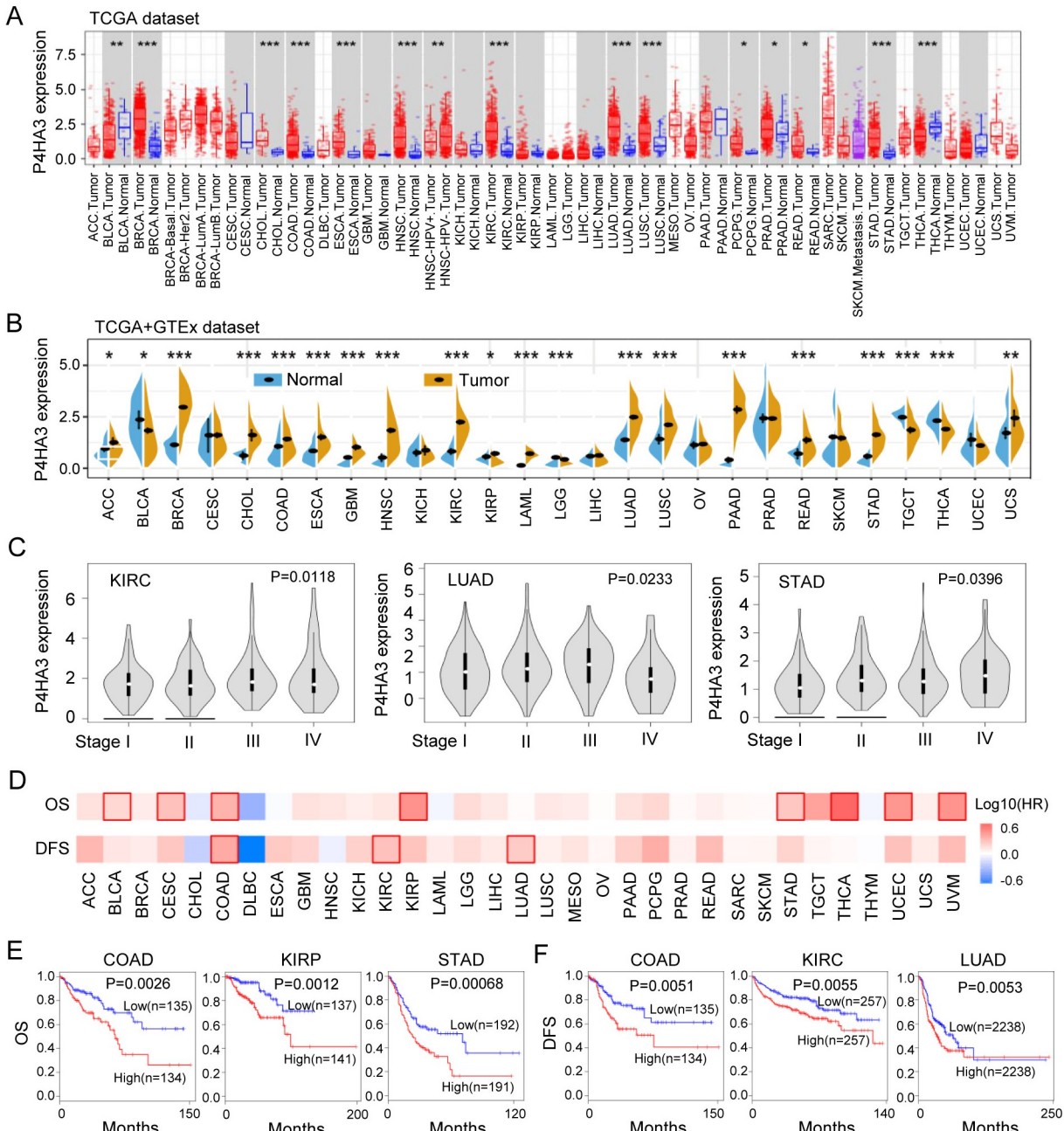

**Fig 1. Expression profiles and prognostic values of P4HA3 in different tumor types from TCGA and GTEx cohorts.** (A) The expression of P4HA3 in different types of cancers or cancer subtypes was analyzed via the TIMER2.0 online resource from TCGA dataset. The red rectangles represent tumor tissue, and the blue rectangles represent normal tissue. *$p < 0.05$; **$p < 0.01$; ***$p < 0.001$. (B) Expression level of P4HA3 was analyzed in different cancer types from TCGA and GTEx data. The brown fusiformis represents tumor tissue, and the red fusiformis represents normal tissue. *$p < 0.05$; **$p < 0.01$; ***$p < 0.001$. (C) P4HA3 expression was positively correlated with advanced stages of KIRC, LUAD and STAD (based on TCGA and GTEx data). (D) OS and DFS analysis of P4HA3 in the 33 types of cancers, OS: Overall survival; DFS: Disease-free survival. (E) OS of P4HA3 in COAD ($p = 0.0026$), KIRP ($p = 0.0012$), STAD ($p = 0.00068$) by Kaplan-Meier survival curves in GEPIA database. (F) DFS of P4HA3 in COAD ($p = 0.0051$), KIRC ($p = 0.0055$), LUAD ($p = 0.0053$) by Kaplan-Meier survival curves in GEPIA database.

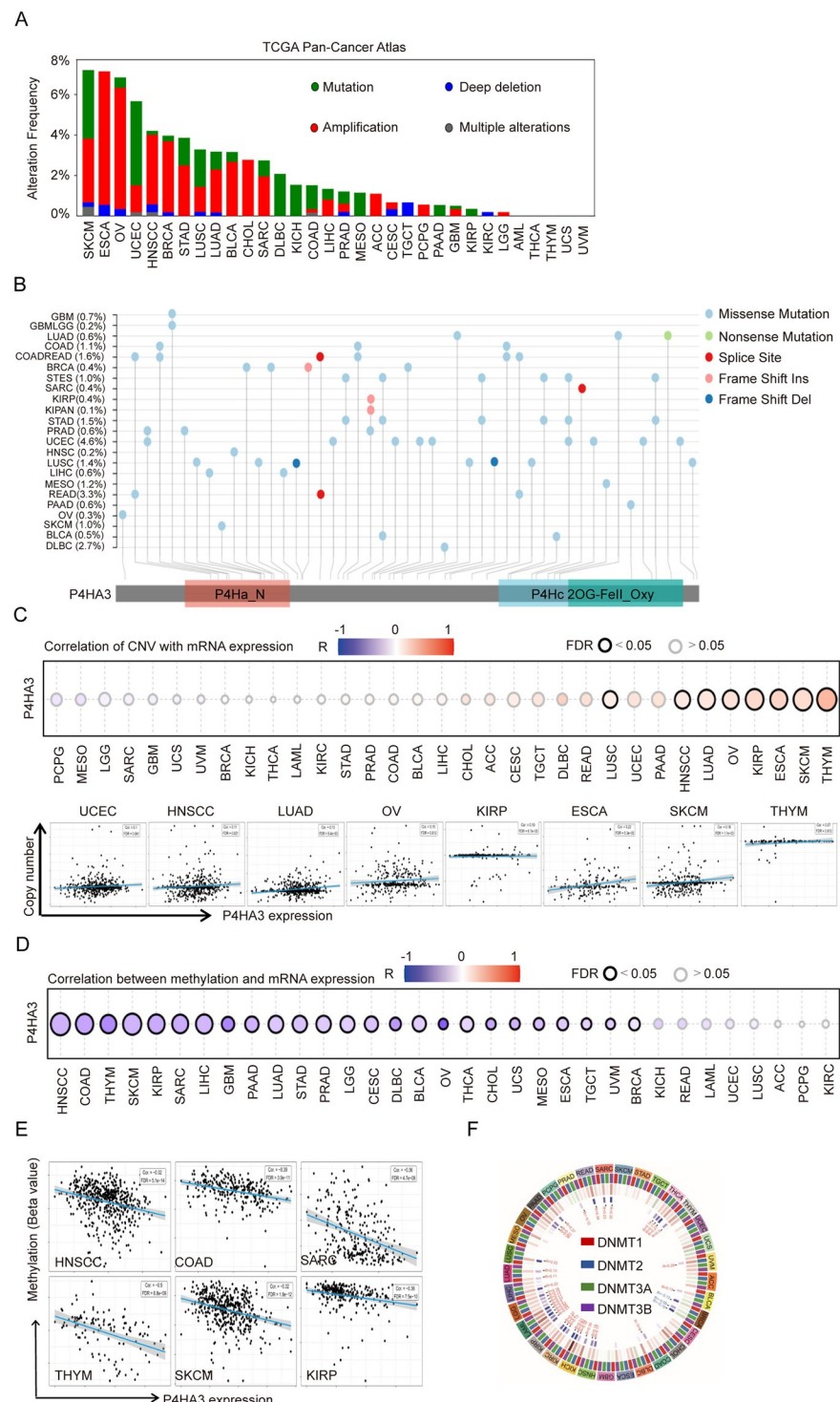

**Fig 2. Genetic and epigenetic alteration of P4HA3 in various cancers.** (A) The genetic alteration type and frequency of P4HA3 were analyzed by using the cBioPortal tool in TCGA tumors dataset. The results were showed as a histogram of the alteration type and frequency of P4HA3 in 32 types of cancers, color images were available online. (B) Mutation types and frequency of P4HA3 in 23 types of cancers, mutation diagram of P4HA3 in different human cancers across protein domains. (C) The association between P4HA3 expression and copy number variation was analyzed via the GSCA tool in 33 types of cancers, especially in LUSC, HNSCC, LUAD, OV, KIRP, ESCA, SKCM, and THYM. (D and E) The association between P4HA3 expression and DNA methylation in 33 types of cancers, especially in HNSCC, COAD, THYM, SKCM, KIRP, and SARC. (F) Correlation between P4HA3 expression and DNA methylation transferase DNMT1, DNMT2, DNMT3A and DNMT3B across 33 types of cancers.

expression level in 33 types of cancers (Fig 2F). Our results showed that P4HA3 upregulation was correlated with DNA copy number amplification and methylation in various of cancers.

## Genome wide association analysis of P4HA3 in different tumor types

To explore the possible biological role of P4HA3 in various of cancers, we used gene co-expression analysis to evaluate P4HA3 function and regulatory mechanism. We found that P4HA3 expression was positively correlated with the typical proliferation proteins, including proliferating cell nuclear antigen (PCNA) and Ki-67 in various of cancers, such as BLCA, BRCA, HNSC, KICH by the heat map ($P < 0.05$) (Fig 3A). In addition, EMT is a crucial step of cancer cell migration and invasion [13], we focused on the possible associations between P4HA3 expression and classic EMT markers, such as Vimentin (VIM), TWIST1, Snail2 (SNAL2), Snail1 (SNAL1), Fibronectin1 (FN1), and N-cadherin (CDH2) [14]. The heat map showed that P4HA3 was significantly positively correlated with the expression of these EMT markers in various of cancers, especially in COAD (Fig 3B). We also analyzed the associations between P4HA3 expression and MMPs (Matrix Metalloproteinase) which also are crucial in cancer metastasis, and found that P4HA3 was significantly positively correlated with the expression of MMPs including Matrix Metalloproteinase 9 (MMP9), Matrix Metalloproteinase 7 (MMP7), Matrix Metalloproteinase 2 (MMP2) and Matrix Metalloproteinase 14 (MMP14), especially MMP2 and MMP14 (Fig 3C). All these data together suggested that high P4HA3 expression was widely associated with cancer proliferation and metastasis.

## Correlation between P4HA3 expression and stromal and immune infiltration in cancers

The vital role of the TME including various immune, stromal and tumor cell types in regulating tumor growth, metastasis and prognosis is now generally understood [15,16]. We explored the possible correlation between P4HA3 expression level and the infiltrating immune and stromal cells by using the ESTIMATE algorithm [17], and found that P4HA3 expression was prominently correlated with immune score, stromal score and ESTIMATE score in various of cancers (Fig 4A). It is notably that P4HA3 expression level is positively correlated with immune score, stromal score and ESTIMATE score in BRCA and COAD (Fig 4A). Previous studies revealed that high immune/stromal/ESTIMATE scores were significantly associated with metastasis and poor prognosis in BRCA and COAD [18], suggesting that P4HA3 as an oncogene via increasing immune and stromal cell infiltration in BRCA and COAD. These data indicate that P4HA3 plays a crucial role in regulating the TME.

In order to further explore the relationship between P4HA3 expression and immune cells, we analyzed the association between P4HA3 expression and infiltration of 28 kinds of immune cells in the TISIDB platform [19]. In various of cancers, P4HA3 expression was positively correlated with infiltrated central memory CD8[+] T cell, central memory CD4[+] T cell, Type 1 T helper cell, regulatory T cell, natural killer cell, macrophage and mast cell (Fig 4B). In the previous studies, infiltrating immune cells play different role and different clinical significance in various of cancers [20]. In COAD, the infiltration of immune cells populations including activated CD8[+] T cell, immature dendritic cell, CD56 bright/dim natural killer cell showed antitumor activity. By contrast, CD8[+] T cell, activated B cell, regulatory T cell, M2 macrophage cell, regulatory dendritic cell and neutrophile cell had the characteristic of cancer-promoting [21, 22]. In our present research, we found that P4HA3 expression was positively correlated with activated B cell, regulatory T cell and macrophage cell infiltration in COAD (Fig 4C). These results possibly indicate that P4HA3 promotes tumor development by regulating activated B cell, regulatory T cell and macrophage cell infiltration in COAD.

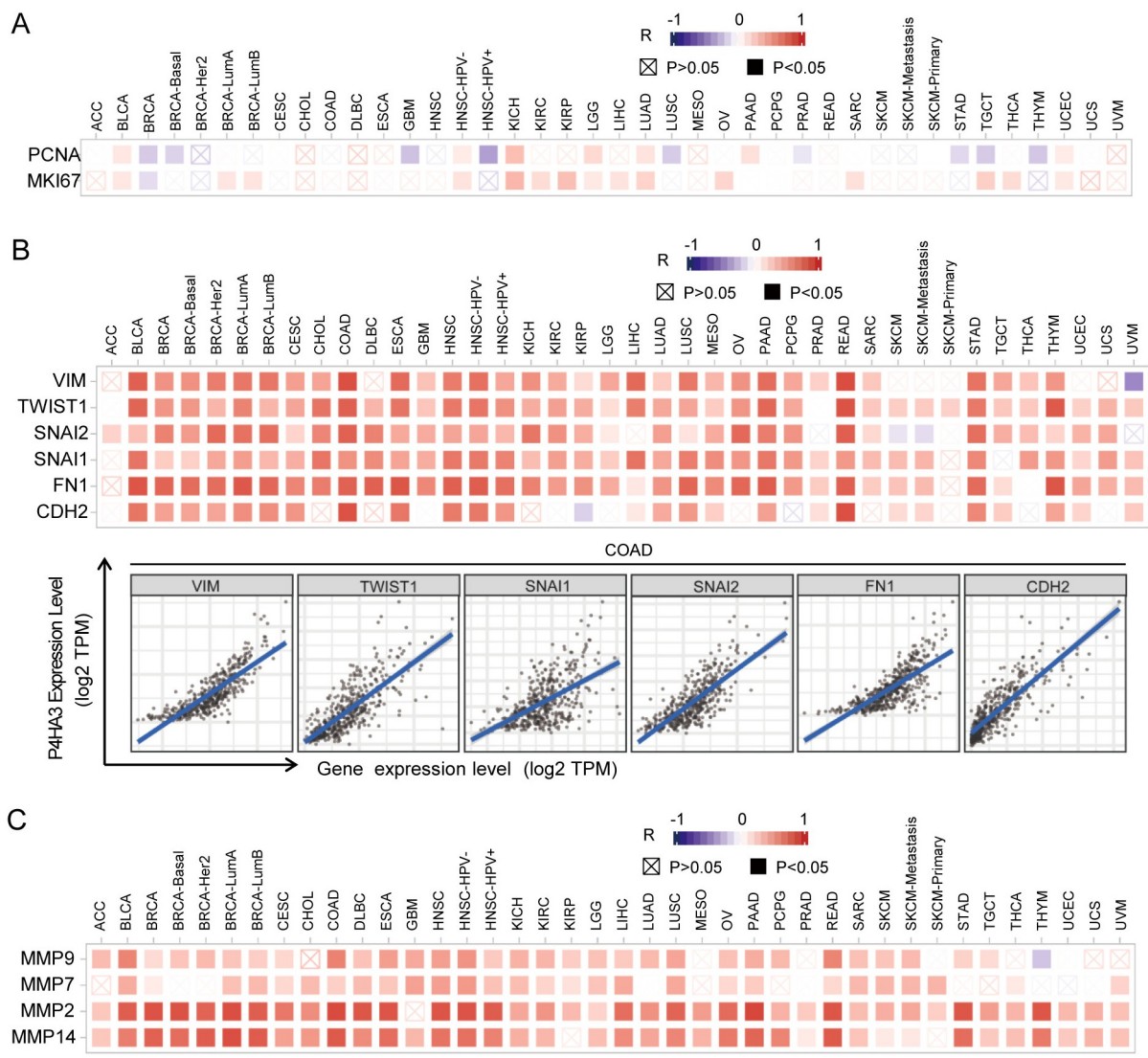

**Fig 3. Correlations of P4HA3 expression and cancer cell proliferation markers and EMT markers across human cancers.** (A) Correlations of P4HA3 and proliferation markers (PCNA and MKI67) were analyzed in different cancer types. (B) Correlations of P4HA3 and EMT markers (Vimentin, TWIST1, Snail1, Snail2, Fibronectin 1, and CDH2) were analyzed in different cancer types, especially in COAD. (C) Correlations analysis on the association of P4HA3 and matrix metalloproteinases (MMP2, MMP7, MMP9 and MMP14).

## Correlation between P4HA3 expression and immune cell markers in cancers

CAFs are closely involved with cancer progression and therapeutic resistance, are the most abundant stromal cells in TME [3]. In order to explore the relationship between P4HA3 expression and infiltrating CAFs, we used four different algorithms (TIDE, XCELL, MCPCOUNTER and EPIC) in our present work. We found that P4HA3 expression level was positively associated with the number of infiltrating CAFs in various of cancers (appeared in at least 3 out 4 algorithms) (Fig 5A). especially in BRCA and COAD, P4HA3 expression level was positively associated with the number of infiltrating CAFs in the 4 algorithms (P < 0.0001) (Fig 5B). Next, we further analyzed the correlation between P4HA3 expression and immune

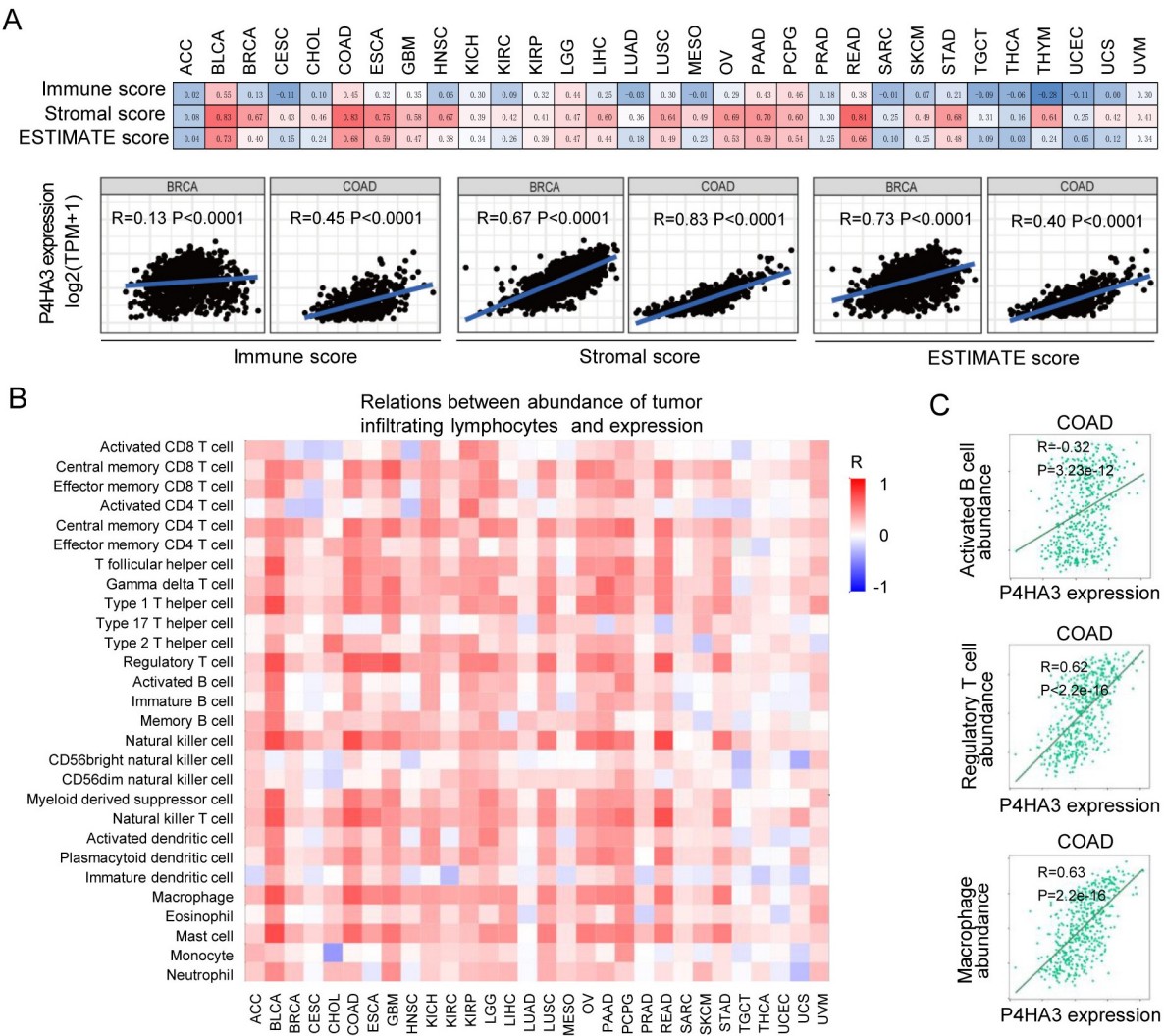

**Fig 4. Correlations of P4HA3 expression with immune infiltration level of tumor micro-environment.** (A) The possible correlations of P4HA3 expression and immune and stromal cells were analyzed by ESTIMATE algorithm, especially in BRCA and COAD. (B) Correlations of P4HA3 expression with 28 types of tumor-infiltrating lymphocytes were showed by using hetmap across 30 types of cancers. (C) Correlations of P4HA3 expression with activated B cells, regulatory T cells and macrophage abundance in COAD.

marker genes, including CSF1R and CD86 (representing monocyte cells), IL10, CD68 and CCL2 (representing tumor-associated macrophage cells; TAM), PTGS2, NOS2 and IRF5 (representing M1 macrophage cells), VSIG4, MS4A4A and CD163 (representing M2 macrophage cells). The results showed that the expression of monocyte, TAM, M1 macrophage and M2 macrophage markers were significantly correlated with P4HA3 expression in BLCA, BRCA, COAD, ESCA, GBM, HNSC, KICH, KIRC, KIRP, LGG, LIHC, LUAD, LUSC, OV, PAAD, PCPG, PRAD, READ, SKCM, STAD, THYM and UVM (P < 0.05) (Fig 5C). Specifically, it was found that CSF1R and CD86 of monocyte were positively correlated with P4HA3 expression in BLCA, COAD, ESCA, READ and THYM; IL10, CD68 and CCL2 of TAMs were positively correlated with P4HA3 expression in BLCA, BRCA, COAD, ESCA, GBM, HNSC, KICH, KIRC, KIRP, LGG, LIHC, LUAD, LUSC, OV, PAAD, PCPG, PRAD, READ, SKCM, STAD, THYM and UVM; PTGS2, NOS2 and IRF5 of M1 phenotype were positively correlated

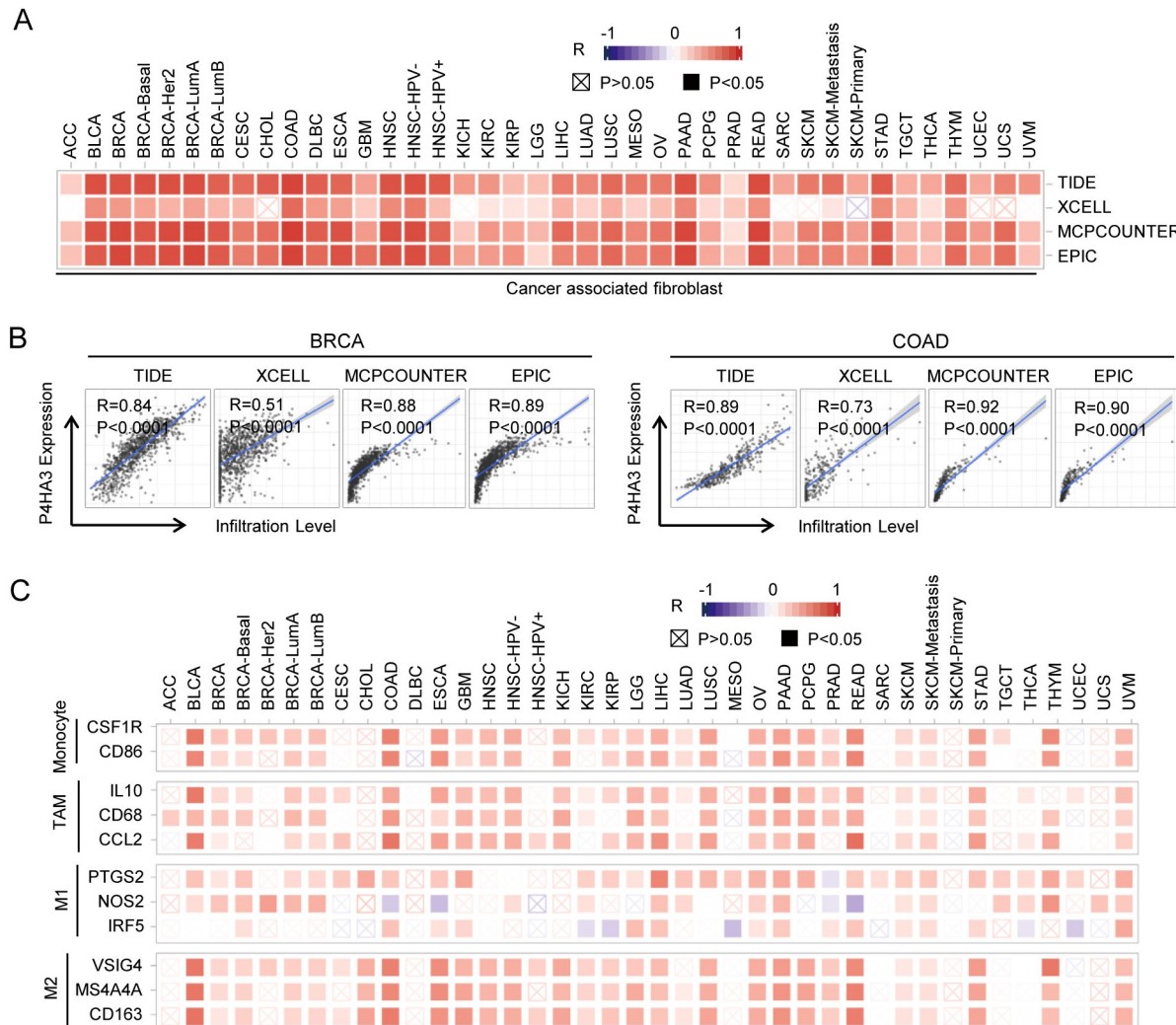

**Fig 5. Correlations of P4HA3 expression with immune cell markers.** (A) Correlations of P4HA3 expression with CAF infiltration were analyzed by different arithmetic TIDE, XCELL, MCPCOUNTER and EPIC in various of cancers. (B) Correlations of P4HA3 expression with CAF infiltration in BRCA and COAD were analyzed by different arithmetic TIDE, XCELL, MCPCOUNTER and EPIC. (C) Correlations of P4HA3 expression with marker genes of monocyte, TAMs, M1 Macrophage and M2 Macrophage in various of cancers.

with P4HA3 expression in BLCA, BRCA, LIHC, LUAD, LUSC, OV, PAAD, SKCM, UVM; VSIG4, MS4A4A and CD163 of M2 phenotype were positively correlated with P4HA3 expression in BLCA, BRCA, CHOL, COAD, ESCA, GBM, HNSC, KICH, KIRC, KIRP, LGG, LIHC, LUSC, OV, PAAD, PCPG, PRAD, READ, SKCM, STAD, THYM and UVM (P < 0.05) (Fig 5C). The role of P4HA3 in regulating TME and cancer progression need more research to confirm.

## Correlations between P4HA3 expression and Microsatellite (MSI) and Tumor mutation burden (TMB) in cancers

P4HA3 was positively correlated with MSI in UCS, COAD, MESO and PRAD, and negatively correlated with MSI in STAD and LUSC (Fig 6A). P4HA3 was positively correlated with TMB

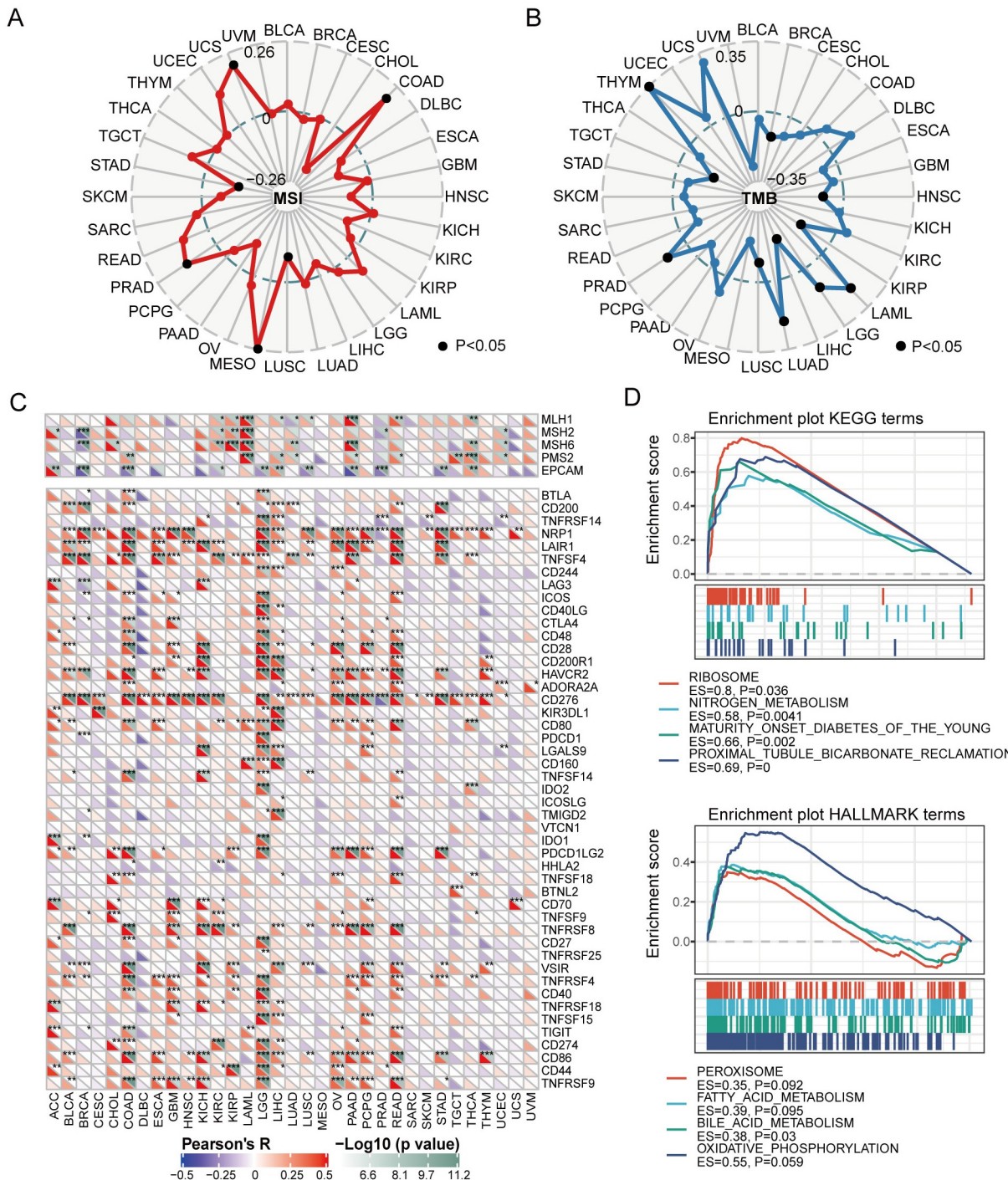

**Fig 6. Correlations of P4HA3 expression with immune checkpoints and other tumor associated characteristics.** (A and B) The associations of P4HA3 expression with MSI and TMB in various of cancers. (C) The correlations of P4HA3 expression with verified immune checkpoints across 33 types of cancers. *$p < 0.05$; **$p < 0.01$; ***$p < 0.001$. (D) GSEA enrichment for P4HA3 expression by KEGG and HALLMARK collections. Each line representing one particular gene cluster with special color.

in THYM, PRAD, LUAD, LGG and LAML, and negatively correlated with TMB in TGCT, LUSC, LIHC, KIRP, HNSC, and BRCA (Fig 6B). In addition, it is very crucial that cancer cells escape the immunosurveillance of host in the prognosis of cancers [23]. We analyzed the correlation between P4HA3 expression and immune checkpoint genes, including PD-1, PD-L1, CD276, and CTLA-4. In BRCA, P4HA3 expression was positively correlated with the expression of CD200, NRP1, LAIR1, TNFSF4, HAVCR2, CD276, VSIR and TNFRSF4, and negatively correlated with the expression of MSH2, MSH6, EPCAM, BTLA, LAG3, ICOS, CTLA-4, PD-L1, TMIGD2, IDO1, TIGIT. In COAD, P4HA3 expression was positively correlated with the expression of PMS2, BTLA, CD200, NRP1, LAIR1, TNFSF4, ICOS, CTLA-4, CD48, CD28, CD200R1, HAVCR2, ADORA2A, CD276, CD80, TNFSF14, PD-1, TNFSF18, TNFRSF8, CD27, VSIR, TNFRSF4, CD40, TIGIT, PD-L1, CD86 and TNFRSF9, and negatively correlated with the EPCAM. In LGG, P4HA3 expression was positively correlated with the expression of BTLA, TNFRSF14, NRP1, LAIR1, TNFSF4, CD244, ICOS, CD40LG, CTLA4, CD48, CD28, CD200R1, HAVCR2, CD276, KIR3DL1, CD80, PDCD1, LGALS9, CD160, TNFSF14, IDO2, IDO1, PDCD1LG2, CD27, TNFRSF25, TNFRSF4, CD40, TNFRSF18, TNFSF15, CD274, CD86, CD44, TNFRSF9 (Fig 6C). Significantly, P4HA3 expression was negatively correlated with the expression of PD-1 in BRCA, and positively correlated with the expression of PD-1 in LGG and PCPG. P4HA3 expression was positively correlated with the expression of PD-L1 in CHOL, COAD, KIRC, LGG, LIHC and PAAD. P4HA3 expression was positively correlated with the expression of CTLA-4 in BLCA, COAD, GBM, LGG, OV and PRAD, and negatively correlated with the expression of CTLA-4 in BRCA (Fig 6C). These results showed that high expression of P4HA3 played a crucial role in cancer progression by regulating immune checkpoint gene, such as PD-1/PD-L1/CTLA-4. Further, we analyzed the functional enrichment of high P4HA3 expression and low P4HA3 expression by GSEA platform. KEGG and Hallmark enrichment plots showed that high expression of P4HA3 was associated with metabolic-related pathways, including metabolism of nitrogen and bile acid (Fig 6D).

## Confirmation by in vitro and in vivo experiments

To confirm the anti-tumor role of P4HA3 deficiency by enhancing anti-tumor immune procedure in cancers, we performed migration, invasion and EdU experiments in vitro and mouse tumor models in vivo. We found that P4HA3 loss significantly inhibited migration, invasion and proliferation abilities in MDA-MB-231, HCT116 and A549 cell lines (Fig 7A and 7B), the related statistical results were shown in S1A and S1B Fig. The knockdown efficiency of P4HA3 in MDA-MB-231, HCT116 and A549 cell lines were detected by using qRT-PCR and western-blot, and the knockdown efficiency was up to 80% (S1C and S1D Fig). In PDX mouse model of TNBC, we found P4HA3-knocked down markedly inhibited PDX tumor growth (Fig 7C and S1E Fig), P4HA3-knocked down also inhibited the expression of P4HA3 and Ki-67 in IHC sections (Fig 7D). More importantly, we found that the volume of the P4HA3-knocked out tumors markedly decreased relative to that of the P4HA3-wildtype tumors after using BMS-1 in c57 mouse model of inoculating 4T1 cells with or without BMS-1, a small-molecular inhibitor of PD-1/PD-L1 (Fig 7E and S1F Fig). We also found P4HA3-knocked out decreased Ki-67 expression and increased PD-1 expression by western-blot assay (Fig 7F). These results indicated higher sensitivity of the P4HA3-knocked out tumor to the PD-1/PD-L1 inhibitor in triple-negative breast cancer.

## P4HA3 expression could reflect the immunotherapy response of TNBC

To determine whether P4HA3 expression could represent the TNBC patients´ immunotherapy response, we retrospectively selected two TNBC patients with neoadjuvant

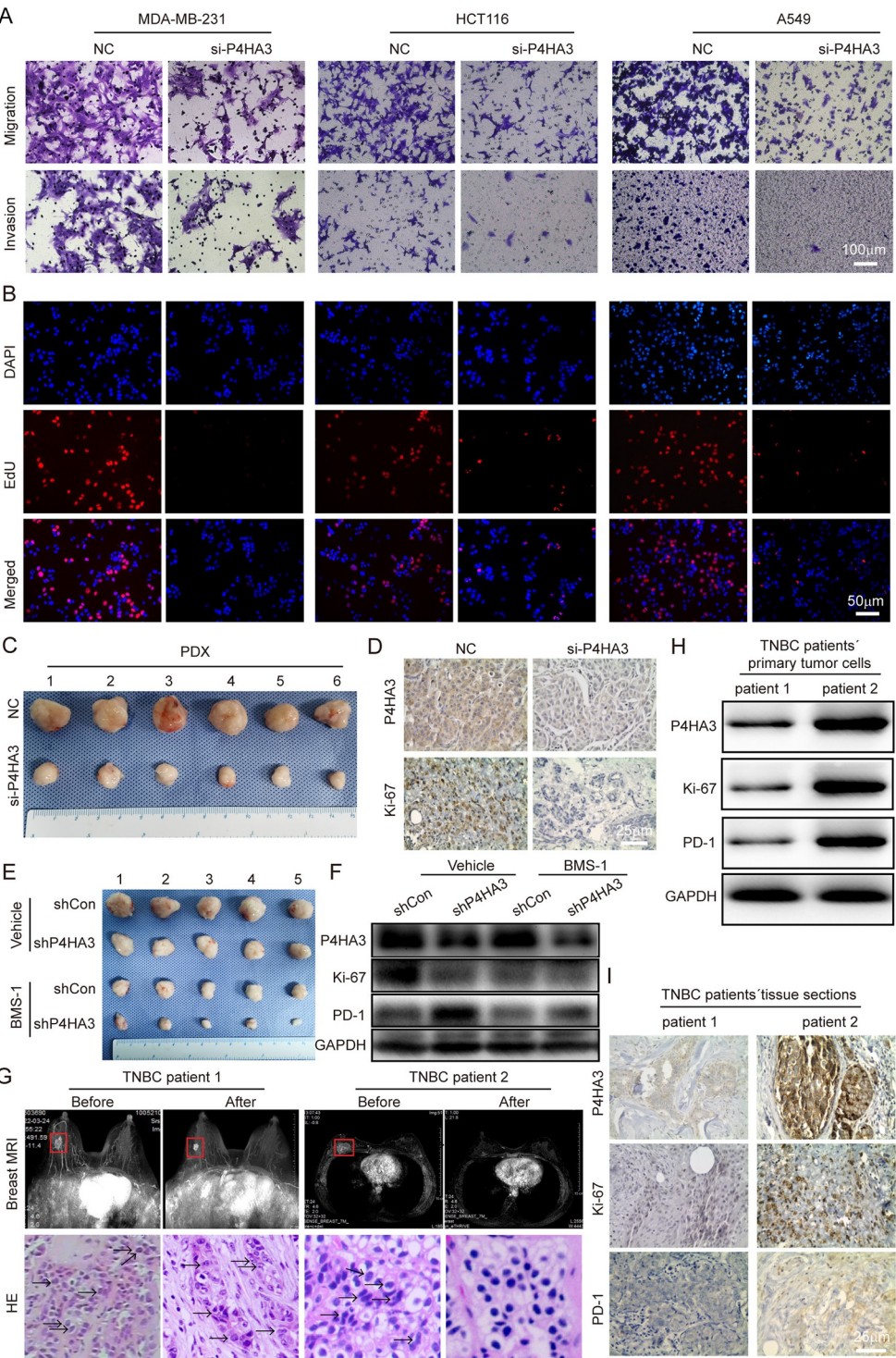

**Fig 7. P4HA3 deficiency increased sensitization to immune checkpoint inhibitors and P4HA3 could predict the clinical therapeutic effect of PD-1 inhibitor (Camrelizumab).** (A) P4HA3 deficiency inhibited migration and invasion abilities of MDA-MB-231, HCT116 and A549 cell lines by transwell assay. Scale bar: 100 μm. (B) P4HA3 deficiency inhibited proliferation ability of MDA-MB-231, HCT116 and A549 cell lines by EdU assay. Scale bar: 50 μm. (C) Representative images of PDXs in the different groups. P4HA3 knockdown by siRNA inhibited tumor growth (n = 5 for each group). (D) PDX tissue sections were subjected to IHC staining against P4HA3 and Ki-67. P4HA3 and Ki-67 were decreased in the knockdown groups by siRNA compared to the NC group. Scale bar: 25 μm. (E) P4HA3-deficient tumors formed by subcutaneous injection of 4T1 cells. ShCon and shP4HA3 tumor-bearing mice

were divided into vehicle groups and BMS-1 groups. The vehicle groups were treated with saline, and BMS-1 groups were treated with 100 U/L PD-1/PD-L1 inhibitor BMS-1 every 3 days for 21 days (n = 5 for each group). Representative tumor images of mice in the different groups. P4HA3 knockdown by siRNA inhibited tumor growth. (F) After sacrifice of the mice, the tumor tissues were separated for WB (western-blot) to detect the expression of P4HA3, Ki-67 and PD-1. Comparisons of P4HA3, Ki-67 and PD-1 expression on different groups in mice are shown. (G) We selected two TNBC patients with neoadjuvant immunotherapy (PD-1 inhibitor: Camrelizumab) to detect the relationship between P4HA3 and immunotherapy therapeutic in TNBC. Breast MRI scan images showed that the tumor size of the two patients was significantly reduced after immunotherapy, and the tumor of second TNBC patient was achieved a pathological complete response (pcr), The red rectangles indicated tumors. The hematoxylin and eosin staining (HE) in needle biopsy of tumors before immunotherapy, the black arrows indicated tumor cells. (H) The tumor tissues from needle biopsy of two TNBC patients before immunotherapy were separated for WB (western-blot) to detect the expression of P4HA3, Ki-67 and PD-1. (I) The tumor tissues from needle biopsy of two TNBC patients before immunotherapy were subjected to IHC staining against P4HA3, Ki-67 and PD-1. Scale bar: 25 μm.

immunotherapy (PD-1 inhibitor: Camrelizumab). Breast MRI scan images showed that the tumor size of the two patients was significantly reduced after immunotherapy, and the tumor of second TNBC patient was totally didn't find. Meanwhile, we used hematoxylin and eosin staining (HE) in needle biopsy or intact tissues of tumors before or after immunotherapy to show immunotherapy effect, we found that the tumor of second TNBC patient was achieved a pathological complete response (pcr) after immunotherapy (Fig 7G). More importantly, the tumor tissues from needle biopsy of two TNBC patients before immunotherapy were conducted for WB and IHC staining to detect the expression of P4HA3, Ki-67 and PD-1, we found that the expression of P4HA3, Ki-67 and PD-1 in second TNBC patient was significantly higher than the first TNBC patient (Fig 7H and 7I). Above results indicated that TNBC patients with high expression of P4HA3 and PD-1 will possibly benefit from immunotherapy, P4HA3 could be a predictor for immunotherapy response of TNBC patients like PD-1.

## Discussion

In the previous study, P4HA3 was aberrantly expressed in gastric cancer [7], colon cancer and squamous cell carcinomas [24]. In our study, the expression of P4HA3 was investigated in the 33 types of tumors in the databases, the expression of P4HA3 was generally upregulated among various of tumors. The biological function of P4HA3 was proved to be involved in EMT, migration and invasion in colon cancer [24]. We integrated analyzed the function of P4HA3 among 33 types of cancers, and found that P4HA3 was associated with proliferation markers (PCNA and MKI67), EMT markers (VIM, TWIST1, SNAI1, SNAI2, FN1 and CDH2), extracellular matrix (ECM) markers (MMP9, MMP7, MMP2 and MMP14). In the study, we found that P4HA3 significantly promoted the migration, invasion and proliferation abilities of MDA-MB-231, HCT116 and A549 cells. According to these results, we conducted that P4HA3 conferred malignant phenotypes to various of cancer cells via activating EMT, in the further, we will perform more experiments to verify this conjecture. Enrichment analysis results indicated that high expression of P4HA3 was associated with metabolic-related pathways, the metabolic disorder of host and cancer cells are crucial for understanding tumor origin and development [25]. In NSCLC, TGF-β pathway altered the amino acid metabolism and induced EMT happen by increasing P4HA3 expression, consequently contributed to cancer cell malignant progression [26]. We supposed that P4HA3 was associated with EMT, ECM malignant phenotypes to various of cancer cells via amino acid metabolism activation, these results provided us some new insights of P4HA3 and amino acid metabolism to search for effective therapeutic targets, biological experiments are needed to verify these new findings and promote clinical transformation.

In this study, we firstly explored the significant role of P4HA3 deficiency in promoting anti-tumor immunity and immunotherapeutic limitation in various of cancers. The rapid

development of cancer immunotherapy has significantly extended the life of tumor patients especially advanced NSCLC. Basing on the successful development of immune checkpoint blockade inhibitors (ICBs) against PD-1 (e.g., pembrolizumab, nivolumab), PD-L1 (e.g., atezo-lizumab, avelumab) and CTLA4 (e.g., ipilimumab), ICBs have produced great responses in many tumors [27]. Various of creative immunotherapies are currently undergoing clinical trial for cancer treatment such as chimeric antigen receptor-T cell immunotherapies (CAR-T) [28]. but for BRCA as an immune-cold (non-T cell infiltration) solid tumor, these patients could not get considerable clinical benefit from ICBs, therefore, the diversity of immune evasion remains a key challenge in converting "cold" tumors into "hot" (T cell infiltration) ones [29]. In the present study, P4HA3 deficiency can strengthen anti-tumor immunity by increasing PD-1 expression and decreasing Ki-67 expression in BRCA. It would be logical that P4HA3 deficiency enhanced susceptibility to ICB therapy for breast cancer. This inference was proved in breast cancer mouse model receiving BMS-1 (PD-1/PD-L1 inhibitor) therapy (Fig 6E). Therefore, these results indicated the potential role of P4HA3 in tumor immunology and prognostic biomarker of cancers.

CAFs are the most abundant and hyper-activated in cancer progression within TME [3]. All in all, CAFs are involved in tumor initiation, angiogenesis, metastasis, chemoresistance and immunosuppression via cell-cell interaction and the production of multiple regulatory molecules [30, 31]. Previous studies showed that CD10$^+$GPR77$^+$ CAF subset provided a survival niche for cancer stem cells and induced drug resistance in breast and lung cancer [32], they found that CCL18-PITPNM3 could induce the information of CD10$^+$GPR77$^+$ CAFs, blocking CCL18-PITPNM3 signal axis could not only reverse chemoresistance and immunosuppression and inhibit cancer progression but also prevent the development of pro-tumor CAFs in the early stage of cancer progression [33]. In our study, we found that the infiltration of CAFs was notably associated with P4HA3 expression in various of tumors, especially BRCA, suggesting that P4HA3 may involve in the generation and evolution of CAFs. In the further, we will conduct more experiments to verify the viewpoint that P4HA3 may be an attractive therapeutic target by blocking pro-tumor CAFs and reversing immunosuppression and chemoresistance in the TME.

## Materials and methods

### Ethics statement

The study protocol and patients' samples were approved by The Ethics Committee of the Zhujiang Hospital, Southern Medical University (Guangzhou, China). All patients provided informed written consent. All clinical tissue specimens from patients were acquired with patients' written informed consents and all experiments were conducted in accordance with the Declaration of Helsinki. The animal experiments were approved by the Animal Care and Use Committee of Zhujiang Hospital, south medical university (Guangzhou, China), the committee number is LAEC-2021-103. The maximal tumor size/burden was not exceeded the permissible tolerance in institutional guidelines.

### P4HA3 expression and survival analysis

The expression of P4HA3 in tumor and adjacent para-cancer tissues, as well as in various tumor stages across 33 types of cancers was analyzed using the TIMER2.0 online tool (http://timer.comp-genomics.org/) [34]. The expression of P4HA3 in normal tissues was obtained from Genotype-Tissue Expression Project (GTEx) [35]. To explore the association between P4HA3 expression and patient survival, the GEPIA2.0 database was used (http://gepia2.

cancerpku.cn/#index) [36]. The median expression of P4HA3 was defined as the cutoff value to determine the high or low expression of P4HA3 by using the "Survival Map" module.

## Genetic and DNA methylation alteration of P4HA3

The mutation frequency of P4HA3 in tumor tissues across 33 types of cancers was present by the cBio Cancer Genomics Portal (http://cbioportal.org) [37]. The specific genetic alteration information of P4HA3 was conducted by the "TCGA Pan-Cancer Atlas Studies" dataset. To evaluate the effect of copy number variation and DNA methylation on P4HA3 expression, the GSCA (Gene Set Cancer Analysis) (http://bioinfo.life.hust.edu.cn/GSCA/#/) was used [38]. The correlation analyses for P4HA3 expression and copy number variation, DNA methylation levels as well as the expression of methylation regulators (DNMT1, DNMT2, DNMT3A, and DNMT3B) across various cancers were conducted.

## Co-expression analysis of P4HA3

We utilized the "Gene_Corr" module of TIMER2.0 to explore the association between the expression of P4HA3 and tumor malignancy and immune related genes in various cancers [34]. Several tumor malignancy and immune related genes were analyzed, including proliferation markers (PCNA and MKI67), epithelial-mesenchymal transition markers (VIM, TWIST1, and CDH2, etc.), microsatellite instability markers (MLH1, MSH2, MSH6, and PMS2, etc.), macrophage markers (CD86, CSF1R, and IL10, etc.), and immune checkpoint markers (PDCD1, CTLA4, and LAG3, etc.). The p-value and correlation coefficient were conducted by the Spearman correlation analysis method. $p < 0.05$ was considered significant.

## Immune microenvironment analysis

In order to understand the potential effect of P4HA3 on immune microenvironment, the stromal/immune/estimate score and immune cell infiltration levels were evaluated. The stromal/immune/estimate score of each tumor sample were evaluated by the R package "estimate" [17]. Spearman correlation analysis was used to reveal the associations between P4HA3 expression and stromal/immune/estimate scores. We used the TISIDB online platform to evaluate the correlation between P4HA3 expression and immune cell abundance [19]. As for cancer-associated fibroblast infiltration, four algorithms (TIDE, XCELL, MCPCOUNTER, and EPIC) were used. The purity-adjusted p-value and correlation coefficient were conducted by the Spearman's test. $p < 0.05$ was considered significant.

## Tumor mutational burden (TMB) and microsatellite instability (MSI) analysis

TMB was defined as the number of mutations (including somatic, coding, base substitution and indel) detected per million genomes. We obtained the TCGA genomic data across 33 types of cancers from the cBio Cancer Genomics Portal and calculated the TMB of each sample [37]. We obtained TCGA MSI score (MANTIS) across 33 types of cancers by "cBioPortalData" the R package [39]. The associations between P4HA3 expression and TMB/MSI scores were calculated by Spearman's test. $p < 0.05$ was considered significant.

## Gene set enrichment analysis (GSEA)

To explore the underlying mechanism of P4HA3, the GSEA was performed, which Kyoto Encyclopedia of Genes and Genomes (KEGG) and Hallmark module of the Molecular Signatures Database were selected [40]. Each of the top four pathway was present.

## Cell migration and invasion assays

In this study, to explore the migration and metastasis abilities of breast cancer, colon cancer and lung cancer cells, we used transwell chambers for building the cell barrier. In invasion assay, a layer of Matrigel (BD Bioscience) was placed above the chamber. Indicated cells were pretreated with siP4HA3 (RiboBio, Guangzhou) for 48 h, and was subsequently trypsinized and washed by PBS. Two thousand cells were seeded into the upper chambers in DMEM with 3% FBS, and the lower chambers were added into DMEM with 10% FBS. About 12 h, the chambers were collected and quantified by photographing in 3 random fields.

## EdU assay

In this study, we used an EdU (5-ethynyl-2´-deoxyuridine; Invitrogen, California, USA) assay to detect the proliferation abilities of breast cancer, colon cancer and lung cancer cells. Indicated cells were pretreated with siP4HA3 (RiboBio, Guangzhou) for 48 h, and $2 \times 10^3$ cells were plated in 96-well plates for 24 h. The cells were incubated with 10 μM EdU solution for 24 h, and sequent fixation, permeabilization and EdU staining. The results were determined with fluorescence microscopy and were collected and quantified by photographing in 3 random fields.

## In vivo experiments

To explore the influence of P4HA3 on TNBC tumorigenesis, we established one case PDX model as follows: TNBC clinical specimens were obtained from TNBC patient (the tumor size was about 3.5 cm, the patient age´s age was 48 years old) who experienced tumor excised at Zhujiang Hospital, south medical university (Guangzhou, China) in 2022. The tumor was cut into small incision and putted into the fourth pair mammary fat pads of anaesthetized four-week-old NOSCID. When the maximum diameter of PDX was up to 5mm about two months, the siRNA against to P4HA3 were intravenous injection of NOSCID one time every 3 days about 6 weeks. Tumor volumes did not exceed the maximum volumes according to the requirement of Animal Care and Use Committee. The tumors were measured every one week and was calculated as the formula: $V = 1/2 \times \text{width}^2 \times \text{length}$.

To explore the relationship between P4HA3 and immunotherapy of TNBC, we established subcutaneous tumor model. In 4T1 cells, we steady knocked out P4HA3 by transducing shCon or shP4HA3 lentivirus selected by using puromycin. $1 \times 10^6$ 4T1 cells with shCon or shP4HA3 were injected into the third right mammary fat pads with C57/BL6 background. Once tumors reached 5 mm in diameter according to the standard modified formula Volume $(\text{mm}^3) = (\text{length} \times \text{height}^2)/2$, half of the shCon and shP4HA3 mice were treated with 100 U/L PD-1/PD-L1 inhibitor BMS-1 (Selleck Cas: No. 1675201-83-8) every 3 days. After 21 days of treatment, the tumors were excised, tumor tissues were individually preserved for IHC, ISH, RNA and protein extraction.

## Statistical analysis

All statistical analyses were conducted by student t-test by using Graph pad prism 5.0, the results in vitro and in vivo were repeated three independent experiments. The statistical analyses of results in vitro and in vivo were separately presented as mean ± SD and mean ± SEM.

## Supporting information

**S1 Fig. P4HA3 deficiency inhibited malignant biological abilities of cancer cells.** The statistical analysis from Fig 7. (A) P4HA3 deficiency inhibited migration and invasion abilities of

MDA-MB-231, HCT116 and A549 cell lines by transwell assay, (***p < 0.001 among the different groups by paired Student´s t tests, mean ± SD). (B) P4HA3 deficiency inhibited proliferation ability of MDA-MB-231, HCT116 and A549 cell lines by EdU assay, (***p < 0.001 among the different groups by paired Student´s t tests, mean ± SD). (C) The knockdown efficiency of P4HA3 in MDA-MB-231, HCT116 and A549 cell lines were detected by using qRT-PCR, (***p < 0.001 among the different groups by paired Student´s t tests, mean ± SD). (D) The knockdown efficiency of P4HA3 in MDA-MB-231, HCT116 and A549 cell lines were detected by using WB. (E) Tumor size in different groups were calculated every 7 days over 2 months. (***p < 0.001 between the NC group and siRNA group by paired Student´s t tests, mean ± SD). (F) Tumor size in different groups were calculated every one week over one month. (***p < 0.001 among the different groups by paired Student´s t tests, mean ± SD). (TIF)

## Author Contributions

**Conceptualization:** Hong Yan Huang, Fu Wei Zhang, Jie Yu.

**Formal analysis:** Hong Yan Huang, Hai Yun Jin, Yong Sheng Huang, Shu Wei Ren.

**Funding acquisition:** Hong Yan Huang, Jie Yu, Yong Sheng Huang.

**Investigation:** Hong Yan Huang, Fu Wei Zhang, Jie Yu.

**Methodology:** Hong Yan Huang, Fu Wei Zhang, Jie Yu, Yan Hong Xiao, Di Zhu, XiaoLin Yi, XiaoHua Lin, Ming Jin.

**Project administration:** Hong Yan Huang, Fu Wei Zhang, Jie Yu.

**Resources:** Hai Yun Jin, Yong Sheng Huang, Shu Wei Ren.

**Software:** Ming Jin, Yong Sheng Huang, Shu Wei Ren.

**Supervision:** Hai Yun Jin, Yong Sheng Huang, Shu Wei Ren.

**Validation:** Hai Yun Jin, Yong Sheng Huang, Shu Wei Ren.

**Visualization:** Hong Yan Huang, Fu Wei Zhang, Jie Yu.

**Writing – original draft:** Hong Yan Huang, Fu Wei Zhang, Jie Yu.

**Writing – review & editing:** Hong Yan Huang, Hai Yun Jin, Yong Sheng Huang, Shu Wei Ren.

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
