## [Decision Letter · Decision Letter 0]

6 Sep 2024

Dear Professor huang,

Thank you very much for submitting your manuscript "A computational analysis of the oncogenic and anti-tumor immunity role of P4HA3 in human cancers" for consideration at PLOS Computational Biology. As with all papers reviewed by the journal, your manuscript was reviewed by members of the editorial board and by several independent reviewers. The reviewers appreciated the attention to an important topic. Based on the reviews, we are likely to accept this manuscript for publication, providing that you modify the manuscript according to the review recommendations.

Sincerely,

Hailin Tang

Guest Editor

PLOS Computational Biology

Ilya Ioshikhes

Section Editor

PLOS Computational Biology

Reviewer's Responses to Questions

**Comments to the Authors:**

Reviewer #1: This manuscript is a remarkable tour-de-force that offers a thorough understanding of the oncogenic and prognostic role of P4HA3 in various cancers. Its principal findings highlight the underappreciated antitumor therapeutic potential of P4HA3, which may serve as a valuable biomarker for predicting the efficacy of immunotherapy in cancer patients. Overall, this is a robust piece of work that could benefit from some minor improvements.

My primary concerns are as follows:

In the functional experiments, the authors concentrated on breast cancer. However, the expression of P4HA3 was not correlated with overall survival in the TCGA breast cancer cohort. How does P4HA3 expression influence the survival of breast cancer patients in the alternative dataset?

The representation of P4HA3 in Figure 7H is somewhat misinterpreted: the protein expression of P4HA3 appears to be elevated in the P4HA3 knockdown group (shP4HA3) compared to the control group. Please verify the data meticulously.

Reviewer #2: Comments to the Author

Huang et al. systematically describe the oncogenic and anti-tumor immunity of P4HA3 in human cancers, which P4HA3 is generally over-expressed in human cancers to drive proliferative and metastasis properties. The current manuscript focuses on the role of P4HA3 in various of cancers and anti-tumor immunotherapy of PD-1/PD-L1 inhibitor in triple-negative breast cancer. The investigators intensively completed P4HA3 expression by using different databases including TIMER2.0, GTEx, GEPIA2.0 and TCGA. Genetic and DNA methylation alterations, survival analysis and proteins co-expression analysis of P4HA3 in cBio Cancer Genomics Portal, TCGA, GSCA and TIMER2.0. The correlation between P4HA3 expression and immune infiltration was analyzed by TIDE, XCELL, MCPCOUNTER, and EPIC. Besides above computational analyses, they performed some molecular experiments to confirm the functional role of P4HA3 in different tumors. More importantly, they again verified anti-tumor immunity of P4HA3 by many clinical data. Using these data, the authors conclude that P4HA3 abnormal expression could be a useful biomarker for predicting the effectiveness of immunotherapy in cancer patients. This study has a strong innovation, but, i found many small problems to be resolved before publication:

Minor points:

1- Three are many grammatical errors throughout the manuscript.

2- Please number your figure and provide the figure legends as it will make it easier to understand your work.

3- The mousseline is incorrectly written as NOSCID instead of NOD-SCID mice.

4- Please provide the patient information of PDX model.

Reviewer #3: This paper assesses the role of P4HA3 in cancer development and comprehensive analyses among human cancers, it is a good idea that the authors find P4HA3 deficiency inhibited the proliferation, migration, and invasion abilities of tumor cells, and promoted anti-tumor immunotherapy of PD-1/PD-L1 inhibitors. I think the paper is interesting. However, what is the application of P4HA3 in the clinical tumor gene panel (such as the MSK-MPACT panel)?

**Have the authors made all data and (if applicable) computational code underlying the findings in their manuscript fully available?**

Reviewer #1: None

Reviewer #2: Yes

Reviewer #3: Yes

PLOS authors have the option to publish the peer review history of their article (what does this mean?). If published, this will include your full peer review and any attached files.

Reviewer #1: No

Reviewer #2: No

Reviewer #3: No

Figure Files:

Data Requirements:

Reproducibility:

References:

---

## [Decision Letter · Decision Letter 1]

9 Oct 2024

Dear Professor huang,

We are pleased to inform you that your manuscript 'A computational analysis of the oncogenic and anti-tumor immunity role of P4HA3 in human cancers' has been provisionally accepted for publication in PLOS Computational Biology.

Best regards,

Hailin Tang

Guest Editor

PLOS Computational Biology

Ilya Ioshikhes

Section Editor

PLOS Computational Biology

Reviewer's Responses to Questions

**Comments to the Authors:**

Reviewer #1: no

Reviewer #2: The author has addressed all my concerns. Thanks.

Reviewer #3: Author answers all questions

**Have the authors made all data and (if applicable) computational code underlying the findings in their manuscript fully available?**

Reviewer #1: Yes

Reviewer #2: Yes

Reviewer #3: None

PLOS authors have the option to publish the peer review history of their article (what does this mean?). If published, this will include your full peer review and any attached files.

Reviewer #1: No

Reviewer #2: No

Reviewer #3: No

---

## [Editor Report · Acceptance letter]

3 Nov 2024

PCOMPBIOL-D-24-01089R1 

A computational analysis of the oncogenic and anti-tumor immunity role of P4HA3 in human cancers

Dear Dr huang,

I am pleased to inform you that your manuscript has been formally accepted for publication in PLOS Computational Biology. Your manuscript is now with our production department and you will be notified of the publication date in due course.

With kind regards,

Dorothy Lannert
